# The Streaming Web-Based Exercise at Home Study for Breast and Prostate Cancer Survivors: A Feasibility Study Protocol

**DOI:** 10.3390/mps6030051

**Published:** 2023-05-17

**Authors:** Celina H. Shirazipour, Rachel M. Ruggieri-Bacani, Laura Lockshon, Christopher Waring, Aubrey Jarman, Novalyn Cruz, Catherine Bresee, Angela J. Fong, Pao-Hwa Lin, Gillian Gresham, Arash Asher, Stephen J. Freedland

**Affiliations:** 1Cedars-Sinai Cancer, Cedars-Sinai Medical Center, Los Angeles, CA 90048, USA; rachel.ruggieri@cshs.org (R.M.R.-B.); christopher.waring@cshs.org (C.W.); aubrey.jarman@cshs.org (A.J.); novalyn.cruz@cshs.org (N.C.); catherine.bresee@cshs.org (C.B.); gillian.gresham@cshs.org (G.G.); arash.asher@cshs.org (A.A.); stephen.freedland@cshs.org (S.J.F.); 2David Geffen School of Medicine, University of California Los Angeles, Los Angeles, CA 90095, USA; 3Section of Behavioral Sciences, Rutgers Cancer Institute of New Jersey, New Brunswick, NJ 08903, USA; angela.fong@rutgers.edu; 4Department of Medicine, Duke University, Durham, NC 27708, USA; pao.hwa.lin@duke.edu; 5Urology Section, Durham VA Medical Center, Durham, NC 27705, USA

**Keywords:** cancer survivor, exercise, exercise therapy, physical activity, telemedicine, telerehabilitation

## Abstract

Background: Despite the known benefits of physical activity in cancer survivors, adherence to exercise guidelines remains low. Known barriers to adhering to guidelines include a lack of time and an unwillingness to return to treatment facilities. Virtual exercise programming could assist in mitigating these barriers. This protocol presents a single arm pilot study exploring the feasibility of personalized Zoom-delivered exercise training for breast and prostate cancer survivors. A secondary objective is to determine the preliminary efficacy of participation on body composition, estimated VO_2max_, hand grip, one repetition maximum leg press, resting heart rate, resting blood pressure, exercise self-efficacy, and intentions to remain active. Methods: Breast (*n* = 10) and prostate (*n* = 10) cancer survivors will participate in a 24-week feasibility study, including (1) 12 weeks of one-on-one virtual personal training with an exercise physiologist (EP) via Zoom, and (2) individual exercise for a 12-week follow-up period using recordings of Zoom sessions for guidance. Physical assessments and surveys will be implemented at baseline, 12 weeks, and at the end of the study (24 weeks from baseline). Conclusions: While virtual exercise programming became popularized during the pandemic, evidence is still required to understand whether it can successfully address barriers and promote participation.

## 1. Introduction

Exercise is safe and beneficial for cancer survivors [1], and is associated with many physical (e.g., lower risk of all-cause mortality, improved physical functioning, improved fitness and strength, decreased fatigue), psychological (e.g., improved quality of life, decreased anxiety and depression), and social (e.g., increased connectedness and social support) benefits [2,3,4]. To achieve these benefits, the American College of Sports Medicine cancer-specific exercise guidelines recommend aiming for 150 min of moderate-intensity or 75 min of vigorous-intensity aerobic activity per week, muscle-strengthening activities at a moderate intensity at least two days a week for major muscle groups, as well as stretching major muscle groups [5,6]. Unfortunately, adherence to guidelines is low. Estimates of guideline adherence among cancer survivors range from 17 to 47% based on method of assessment, demographics, cancer, type of cancer treatment they are receiving, and prior movement habits (physical activity and sedentary behavior) before diagnoses [4,7,8,9,10,11,12]. Both breast and prostate cancers have high 5-year survival rates, yet they are also common diagnoses [13]. This suggests that there is a growing number of survivors who are living with the effects of a cancer diagnosis and cancer treatment. Thus, they are a target population in need of intervention, and there is a critical need to determine successful approaches for promoting adoption and adherence to exercise programs.

Researchers have sought to identify exercise preferences among survivors to ameliorate low adherence to exercise guidelines [14]. Evidence suggests that cancer survivors would prefer to receive exercise information from fitness experts associated with a cancer center, to be active at home, and to receive individually tailored activities [14]. In addition, cancer survivors also report many environmental barriers to participating in exercise programs, including weather conditions, lack of time, and lack of access to exercise facilities [15]. Evidence suggests that supervised exercise interventions yield greater physical and psychosocial outcomes, and improved adherence compared to non-supervised programs [16]. However, many evidence-based cancer-specific exercise programs did not account for survivor preferences, nor did they address environmental barriers. Technology can be used to address this gap. The potential benefit of technology is that it allows survivors to stay at home and still receive supervised training. Virtual program delivery could also address some of the barriers by eliminating those related to poor weather conditions, as the exercises program would be indoors, and would cut out commute time to exercise sessions. Barriers related to lack of exercise facility access could also be mitigated with virtual program delivery because exercise could be completed anywhere. While many exercise programs transitioned to virtual formats during the COVID-19 pandemic, and have shown to have positive effects on cardiac health, fatigue, and quality of life, it is unclear if virtual programs are feasible to implement and if such programs promote exercise adherence [17,18].

The purpose of this study is to assess the delivery of live virtual personal training exercise sessions among breast and prostate cancer survivors. Breast and prostate cancer survivors were chosen for optimal representation of female and male participants. Breast and prostate cancer are the most common female and male cancers, respectively [19]. Furthermore, they are two of the cancers with the longest natural history, therefore creating a large pool of survivors [13]. Finally, there are similarities between both cancers, such as their being hormonally regulated and, thus, often treated with endocrine therapy, which can impair physical functioning [20].

The primary endpoint is to assess adherence to intervention delivery by measuring attendance. Attendance is defined as frequency of Zoom sessions attended and completed over the 12-week study period. Secondary outcomes include retention, defined as participation from the baseline through to the final assessment; enrollment, defined as the percentage of patients approached who sign consent; Zoom session sound and visual clarity will be measured through end of session survey questions; changes in body composition will be measured by a bioelectrical impedance analysis, hand grip strength, one repetition maximum leg press, estimated VO_2max_, resting heart rate, and resting blood pressure; and indicators of physical activity behavior, which will be measured through survey items assessing exercise self-efficacy and intentions to remain active. An exploratory qualitative objective will focus on understanding the best practices for quality virtual exercise program delivery through exit interviews.

## 2. Experimental Design

### 2.1. Study Design and Participants

This single-arm pilot study will use a pre- and post-test study design. In total, 20 participants will be recruited, 10 female breast cancer survivors and 10 male prostate cancer survivors. Participants will train with an exercise physiologist (EP) for 12 weeks, 3 times per week for 50 min via Zoom (Part 1). For the following 12 weeks (Part 2), participants will be instructed to complete self-directed exercise training guided by 15 recordings from their supervised Zoom sessions chosen by the EP and shared via a link to HIPAA-compliant Box cloud folders. Assessments will take place at baseline, at the end of Part 1 (approximately 12 weeks after baseline/mid-study), and at the end of Part 2 (approximately 24 weeks after starting intervention/end of study). Assessments will include body composition, strength, estimated VO_2max_ and vitals, and surveys assessing indicators of physical activity behavior. Participation will also include an exit interview at the end of the study. Each patient’s active participation will last approximately 6 months. The study timeline is described in Figure 1.

### 2.2. Recruitment and Consent

Participants will be recruited in clinics by participating health care providers (A.A., S.J.F.) or by the study-assigned clinical research coordinator (L.L., C.W.). Assessments to determine eligibility for this study will be completed only after obtaining informed consent. All screening procedures will be performed within 30 days prior to enrollment. If referred by a health care provider, the clinical research coordinator will call the patient and obtain verification of the potential subject’s identity by collecting at least two Electronic Health Record (EHR) identifiers (name, date of birth, social security number, address, email address, etc.). After receiving this information, the clinical research coordinator will conduct a review of comorbidities and oncologic medical history via chart abstraction.Inclusion and exclusion criteria is determined through electronic health records. Full inclusion and exclusion criteria are highlighted in Table 1. To screen for eligibility, potential participants will complete two surveys, either on paper (in clinic) or online through a REDCap link (at home). A pregnancy test will be administered only for women of child-bearing potential. The study physicians will determine safety clearance for participation, either via chart review (if the patient has been seen by physician within the previous three months) or by a phone call and record review (if the patient has not been seen within the previous three months).

### 2.3. Sample Size

Twenty participants will be recruited. This study was designed to primarily assess the number of Zoom sessions that subjects participate in, with a goal to achieve a minimum of 70% of exercise sessions (26 out of 36 sessions). Assuming a null hypotheses of 26 sessions (the minimum number of sessions required to attend over 12 weeks), and a standard deviation of 8, we will have 89% power to detect a significant difference in a one-sample *t*-test at the 0.05 significance level. In estimating percentages of retention and consent, the maximum width of the 95% confidence interval would be 0.46.

### 2.4. Devices

#### 2.4.1. Zoom

Zoom will be used for the EP to deliver the live personal training sessions to participants during Part 1. A secure HIPAA-compliant link will be emailed to the participant prior to the exercise sessions. When the participant clicks on the link, a video call with the EP will start on their device. The participant may need to download software at the beginning of their first exercise session if they have not previously used Zoom on that device. Participants will not need to create an account with Zoom. A Zoom connection test will be scheduled prior to the first exercise session. During this connection test, the study participant and EP will have a Zoom call to test audio and video setup and connection.

During exercise sessions, the EP will record their screen and will save the recording to a Box cloud drive. These recordings will include audio and visual components for both the EP and the participant. Recordings will be password protected. At the start of Part 2, the EP will select 15 recordings that represent varied types of exercises to be shared with the participant. Study staff will provide the participant with a USB drive, or give them access to a Box drive online that contains these selected recordings of sessions performed during Part 1. If using the USB drive, it will be encrypted and the participants will be given the password. If using Box, participants will be emailed a secure HIPAA-compliant link to the cloud drive containing their videos, which will also be password protected.

#### 2.4.2. Fitbit

Patients will be asked to wear a study-provided Fitbit wrist-worn activity monitor for the duration of the intervention for the purpose of tracking their daily activity and sleep over the course of the study (Parts 1 and 2). Patients will be asked to install the Fitbit application on their mobile device and use a patient-specific, study-created HIPAA-compliant Fitbit account. Study staff will assist participants in setting up their Fitbit during their baseline visit. Once Fitbit data is synced to the Fitbit, data will be stored in and downloaded from the Fitbit and monitored via Fitabase (Fitbit data collection and management for research projects). Clinical research coordinators will download activity data following the mid-study and end of study visits. Throughout the study, the EP will remind the participant to wear the Fitbit and to sync their device after each exercise session. If the Fitbit is not synced after reminders, the clinical research coordinator will contact the participant and ask them to sync it. During Part 2 of the intervention, the clinical research coordinator will remind the participants to sync, wear, and charge their Fitbit.

## 3. Procedures

### 3.1. Intervention

The participant and the EP will meet via Zoom three times every week during Part 1 (EP-directed exercise). The participant can choose to have their three weekly sessions on any weekday. Sessions are allowed to be completed on consecutive days. These sessions will include a 5 min warm-up, 40 min moderate intensity exercise consisting of aerobic “cardio spurts” (jog in place, high knees, etc.), chair exercises, resistance training with study provided resistance bands, and a 5 min cool down. The 40 min of moderate intensity exercise will consist of 10 exercises continued for 1 min, each with 15 s of rest between exercises. The intervention will include exercises that target the muscle groups back/biceps, chest/shoulders/triceps, legs, abdominals, and overall aerobics. After the 10 exercises are completed, the participants will be given 2 min of rest, before repeating the 10 exercises 2 more times, totaling 3 rounds. The total weekly exercise sessions will result in 150 min of activity. The EP will rotate between different exercises for each participant to work different muscle groups at different sessions. The EP will personalize the exercises to the patient’s fitness and ability level to maintain a moderate intensity level. Participants will be provided with exercise Therabands of various strengths at the baseline visit, and will gradually move up in band resistance strength throughout the study. The yellow Theraband will be the lightest (1.3 kg of resistance at 100% elongation), then red (1.7 kg of resistance at 100% elongation), green (2.1 kg of resistance at 100% elongation), and blue (2.6 kg of resistance at 100% elongation). Participants will start at different Theraband strengths based on their baselines assessments, or even double up the Therabands if more of a challenge is necessary. The participants will be expected to use a four-legged sturdy chair at their home (i.e., no wheels, solid back with no option to recline) to complete chair-based exercises. Participants will be expected to wear exercise-appropriate clothing. Participants will be expected to have a clear 5 × 6-foot space available to exercise in. Participants may use a mat during the exercise, but it is not required. The intervention is tailored to each participant’s physical fitness and ability levels. During Part 2 (participant-directed exercise), participants will have access to videos of past training sessions with the EP, but no training sessions with the EP will occur. Then, 10–15 of the best videos will be chosen by the EP that will best support the participant in moving forward with self-directed exercise. Figure 2 provides an example of one week of exercise sessions.

### 3.2. Safety and Adverse Events

Adverse events will be documented and reported in a timely manner to ensure the safety of subjects enrolled in the study. Adverse events (muscle soreness, muscle pain, fatigue, etc.) will be documented for each exercise session and addressed immediately. Additional questions about adverse events will also be reviewed and determined if related to the study by an MD investigator following the mid-study and end of study visits. If a participant experiences an injury, issue, or adverse event and remains in the study during Part 1, the EP will tailor their exercise regimen so as not to exacerbate the injury or illness. If a participant becomes injured during Part 2, they will be instructed to self-tailor the exercises to fit their abilities, and to prioritize their healthcare providers’ advice regarding exercise.

### 3.3. Withdrawal

Patients may skip no more than 30% (no more than 10 out of 36 sessions) of the expected exercise sessions after starting Part 1. If a participant misses more than 30% of sessions, the participant will be removed from the study. Participants who miss more than 2 weeks of exercise sessions in a row after starting Part 1 (6 sessions in a row) will be removed from the study. Detraining can be seen as early as two weeks after not participating in exercise, which is the rationale behind participant removal after two consecutive weeks of missed sessions [21,22]. Patients will be asked to complete an end of study visit when removed from the protocol. If participants go on vacation during Part 1, the EP will encourage the participant to continue study participation while on vacation, which is possible given the virtual nature of the training. Participants will be allowed to make up sessions missed due to national holidays or due to cancellations on the part of study staff (e.g., EP sickness, etc.). The make-up sessions will be added on after the 12 weeks of sessions in Part 1. Each participant may have no more than three make-up sessions. If the participant chooses not to make up these sessions, the sessions will count as missed. Make-up sessions will not impact the timing of the mid-study visit. Participants in Part 2 will self-direct their exercise. Participants will not be removed from the study during Part 2 for failure to exercise.

Patients can be taken off the study treatment and/or study at any time at their own request, or they may be withdrawn at the discretion of an investigator for safety, behavioral, or administrative reasons. The reason(s) for discontinuation will be documented and may include voluntarily patient withdraws (follow-up permitted); patient withdraws consent (termination of treatment and follow-up); patient is unable to comply with protocol requirements; patient sustains a major injury or is diagnosed with a new major disease that makes continuation of the protocol unsafe; treating or study physician determines continued participation in the study would not be in the patient’s best interest; patient sustains an injury or develops an issue that makes them unable to be independently ambulatory (patient requires a walker, cane, or other assistance to ambulate); (Part 1 only) the patient skips 30% (more than 10 out of 36 sessions) of the expected exercise sessions, or participant misses more than two weeks’ exercise sessions in a row (6 sessions in a row). Sessions skipped due to a national holiday or cancellation by study staff (a maximum of 3 sessions) will be made up at the end Part 1 and will not count toward this total. Patients will be removed from the study if any of the criteria listed apply. The primary investigator will be notified and will document the reason for study removal and the date the patient was removed on the Case Report Form. The patient will be asked to return for an end of study visit. Patients who are removed from the study or who withdraw during Part 1 may complete a final discussion with the EP regarding personalized exercise recommendations and general physical activity goals. Subjects who withdraw from the study treatment prior to starting study intervention will be replaced.

### 3.4. Data Collection and Measures

All surveys will be completed on paper or online via REDCap, according to participant preference. All physical assessments will be completed in-person during a study appointment at the host cancer center. Baseline in-person visits may be completed at any point between determination of eligibility and first exercise session with the EP. All baseline procedures must be completed within 14 days prior to the first exercise session with EP. Mid-study visits will be completed up to 10 days following the last exercise session with EP, and end of study visits will be completed at 24 weeks *±* 14 days from first exercise session with EP. All concomitant medications and supplements will be collected through medical record review at the baseline, mid-study, and end of study visits.

### 3.5. Screening

A demographics questionnaire will include six questions regarding race, education, employment, household income, and how many people live in the household. Demographics will be collected from charts and via demographics questionnaires. The Physical Activity Readiness Questionnaire (PAR-Q+) will be used for participants to self-report exercise readiness and safety during screening. The PAR-Q+ is a reliable and valid questionnaire that will be administered and completed on paper or online via REDCap [23,24]. Participants who answer “yes” to any of the seven questions regarding pre-existing health conditions will be excluded from this study, unless responses and health records are reviewed by a study physician and further medical clearance is provided. The Godin–Shephard Leisure-Time Exercise Questionnaire (GLTEQ) will be used for participants to self-report activity levels. The GLTEQ will be administered and completed on paper in the clinic, at home, or online via REDCap when subjects are screened for eligibility, and once again at the baseline visit. The GLTEQ asks questions about how active a person is and is a reliable and valid questionnaire for participants to self-report physical activity [25,26]. Cut-offs exist to classify participants as inactive, moderately active, or fully active, which have been validated in cancer populations. Participants who score greater than 23 on the GLTEQ will be excluded from this study, as they would already be meeting the recommended physical activity guidelines.

### 3.6. Primary Outcomes

The primary outcome will assess feasibility through adherence. Attendance will be used as the primary outcome for statistical purposes and power calculations, but retention, enrollment, and manipulation checks will be considered as well. Adherence will be measured through attendance, which is defined as frequency of the 36 Zoom exercise sessions attended and fully completed over the 12-week study time period. A fully completed session is defined as participation of the full exercise session.

### 3.7. Secondary Outcomes

Retention will be measured as a secondary outcome by the percentage of patients approached who sign consent. Retention is defined as participation from baseline through to the final assessment. Enrollment will be measured as a secondary outcome and is defined as the percentage of patients approached who sign consent. A final measurement consists of a manipulation check, as well as sound and visual clarity checks for Zoom sessions. After each session, the EP will ask the participant to measure sound and visual clarity using a scale of 0–100%. A score of 0% would be the worst sound and visual clarity, while 100% would be the best clarity.

#### 3.7.1. Fitness Outcomes

Fitness outcomes are to be presented in the order in which they are collected. Heart rate will be measured during the baseline, mid-study, and end of study visit. Heart rate (HR) will be obtained after the participant has been sitting for at least 5 min [27]. Heart rate will be measured through the use of a pulse oximeter (CMS50D Fingertip Pulse Oximeter, Hong Kong SAR, China). Blood pressure (BP) will be obtained after the participant has been sitting for at least 5 min [27]. It will be measured manually by an EP at the brachial artery using a stethoscope (3M Littman Classic III, 3M Littman, USA) and an adult sized 29–42 cm sphygmomanometer (Durable two-piece blood pressure cuff, Welch Allyn, Skaneateles, NY, USA). The EP will ensure that the participants’ feet are flat on the floor, and legs will not be crossed while measuring blood pressure [27]. A non-invasive bio-electrical impedance scale body will analyze and measures weight, skeletal muscle mass, body mass index (BMI), and body fat percent (FitScan BC-601FS, Tanita, Tokyo, Japan). Participants will stand on the scale with their feet on sensors, while holding hand sensors for approximately 30 s to gather said data. Hand grip tests will be implemented to measure grip strength. The purpose of this test is to measure the maximum isometric strength of the hand and forearm muscles. The handle of the dynamometer (Hydraulic Hand Dynamometer 5030J1, Jamar) will be aligned with the heel of the participants’ palm, and the middle of the forefingers [28]. The participant will be instructed to squeeze as hard as possible for 3–5 s while seated with their elbow bent at a 90-degree angle [29]. The test will be performed two times on each hand regardless of dominance [27]. A one repetition max leg press will be administered as well. The purpose of the test is to measure lower body strength. This test involves the participant using a leg press machine at different weights to find the most weight in five trials at which they can complete one leg press (LD-3 Leg Press/Calf Raise, Bacta Fitness System, Raleigh, NC, USA). The chair of the leg press will be adjusted to fit each participant’s height. Legs will be placed on the plate of the leg press, in front of the participants, and will be bent at a 90-degree angle. A modified Bruce submaximal treadmill test will be used to assess VO_2_ max during the baseline, mid-study, and end of study visit. The test involves slowly increasing treadmill (T655MS, SportsArt Fitness, Mukilteo, WA, USA) incline grade to 15% and speed to 7 MPH to assess cardiorespiratory responses, including HR, BP, and perceived exertion. A maximum of 15% incline grade will be used due to the treadmill’s capability of only being able to lift to a 15% incline. This modifies the Bruce submaximal protocol which usually calls for a maximum of 26% incline grade. The rating of perceived exertion on a 1–10 scale will be measured every minute. HR will be monitored every minute using a pulse oximeter. BP will be measured every 3 min. Each stage of the Bruce submaximal lasts 3 min each, unless the participants’ heart rate is not within 6 beats of the previous minute. If the HR is not within range, the stage will be continued for another minute. During each stage, the speed and incline of the treadmill will increase until the participant cannot continue, or until the participants’ reaches 85% of their maximum HR.

#### 3.7.2. Patient-Reported Outcomes

The Functional Assessment of Cancer Therapy—General (FACT-G) version 4 will be administered during the baseline, mid-study, and end of study visits. The 27 questions in FACT-G measure physical, social, emotional, and functional well-being in cancer patients. Items are assessed on a 5-point scale Likert-type scale. It has been demonstrated that participants have a better quality of life if they score higher on the FACT-G. The FACT-G has been shown to have high coefficients of reliability and validity for patients with mixed cancer diagnoses who are 18 years or older [30,31,32]. To gain an understanding of whether participation is influencing behavior change, participants will be asked to complete a series of questions assessing proximal psychosocial determinants of exercise behavior (self-efficacy, intentions) at each time-point. This will be assessed through 13 questions measured on a 7-point scale. A sample question includes “If you were really motivated and had all the resources that you needed, how confident are you in your ability to engage in exercise three days a week?” The patient would rate their confidence on a scale from 1 (not at all confident) to 7 (very confident). A semi-structured exit interview will be conducted by the principal investigator via video, phone, or in person up to 30 days after the end of study visit. The exit interview will be approximately 45 min to 1 h long. The discussion will focus on participant experiences with virtual exercise instruction and suggestions for how to improve virtual exercise delivery and study experiences in the future. Audio will be recorded on Zoom and uploaded to Box. Sample questions include “What do you think about the exercises you were asked to complete?” and “What could we improve?”.

### 3.8. Statistical Analysis

For all quantitative data collected, means and standard deviations will be computed for continuous measures and proportions for count data. The average number of Zoom sessions attended will be tested in a one-sample T-test against the null hypothesis of 26, which is the minimum number of sessions that need to be attended over 12 weeks to maintain 70% of sessions completed. All quantitative data (Fitbit data, questionnaires, clinical measurements) will be tested for changes over time with mixed-model regression with random subject-intercept, and patient sex as a covariate. All point estimates of change over time will be computed with 95% confidence intervals. All consented patients will be considered in the analysis.

Qualitative interview data will be analyzed using thematic analysis with a focus on understanding participant experiences and how to improve the intervention for future delivery [33]. The rigor of the qualitative data collection and analysis will be guided by a list of questions developed to assess thematic analysis research quality [34].

## 4. Expected Results

For the primary outcomes, we anticipate a strong adherence of 70% (26 out of 36 sessions) for Part 1, the supervised exercise sessions. During Part 2, the participant-directed exercise, we anticipate that adherence to exercise sessions may decrease.

For the secondary outcomes, we anticipate improvements at 3 months, and maintenance of improvements at 6 months.

Given the nature of qualitative research, no a priori hypotheses were developed for the qualitative aspects of the study.

## Figures and Tables

**Figure 1 mps-06-00051-f001:**
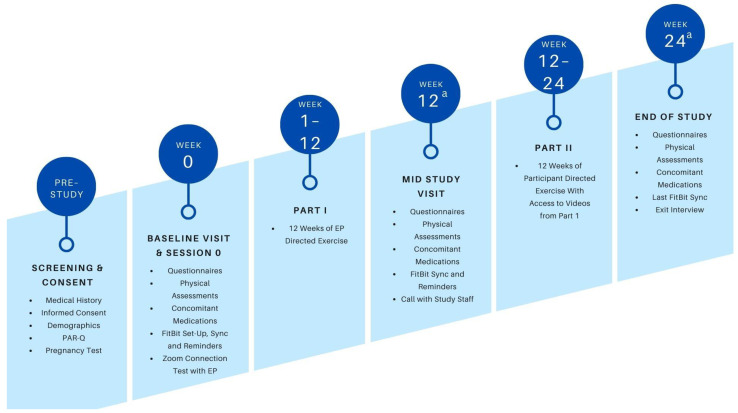
Study timeline. ^a^ Assessments can occur *±* 14 days from date.

**Figure 2 mps-06-00051-f002:**
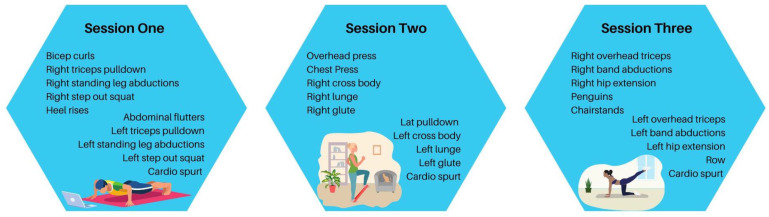
Exercise examples for one week of sessions.

**Table 1 mps-06-00051-t001:** Inclusion and exclusion criteria.

Inclusion Criteria	Exclusion Criteria
Males previously diagnosed with prostate cancer and females previously diagnosed with breast cancer. Minimum of 3 months post-active treatment (Active treatment includes chemotherapy, biologic therapy, radiation therapy, surgery, and any combination). Long-term hormonal/biologic treatments are acceptable. Prostate cancer survivors: currently on active surveillance even if they have not received prior anti-cancer treatment. 18 years of age or older. Able to read, write, and understand English. Ambulatory and physically able to complete all assessments (modified Bruce submaximal treadmill test, leg strength test, grip strength test, and body composition analysis). Access to a solid back chair that does not recline or have wheels. Access to a laptop, tablet, or desktop measuring at least 13 inches across. Access to. A clear 5 × 6-foot space at home in which to exercise. Receives clearance from study physician. Written informed consent obtained from participant. Ability to comply with requirements of the study.	Active treatment planned within the next 6 months (Active treatment includes chemotherapy, biologic therapy, radiation therapy, surgery, and any combination). Long-term hormonal/biologic treatments are acceptable, except for androgen receptor targeted therapies for prostate cancer. Known metastatic disease, grade 3 or higher peripheral neuropathy. Major surgery within 3 months of baseline visit. Positive pregnancy test for women of childbearing potential before start of study. Known allergy to Fitbit device. Answering yes to any question on the Physical Activity Readiness Questionnaire (PAR-Q+), unless cleared by a study physician. Currently meeting physical activity guidelines (score > 23 on Godin–Shephard Leisure-Time Physical Activity Questionnaire). There is no exclusion regarding the type of surgery participants receive as treatment.

## Data Availability

Data sharing is not applicable. No new data were created or analyzed in this study.

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
