# Peer review of "The Streaming Web-Based Exercise at Home Study for Breast and Prostate Cancer Survivors: A Feasibility Study Protocol"

_mps, 2023, doi:10.3390/mps6030051_

Round 1
Reviewer 1 Report
The title is misleading, as well as the entire article. The authors do not produce results about the feasibility of the study protocol proposed, but they explained in detail the procedure to perform a streaming web-based exercise performed at home, adapted to cancer patients. This information, especially after Covid-19 pandemic are very interesting and it is important to share this type of knowledge. I suggest to rewrite the introduction and adjust the objectives in order to this comment. It is clear that this is an ongoing study, but the aim should be the innovative protocol not the effect on these patients or the compliance, because there are no data yet and, in my opinion, it is not in line with the topic of the journal.
Introduction:
Please include some background about the effect of virtual exercise on cancer patients ( i.e. Grazioli, E.; Cerulli, C.; Dimauro, I.; Moretti, E.; Murri, A.; Parisi, A. New Strategy of Home-Based Exercise during Pandemic COVID-19 in Breast Cancer Patients: A Case Study. Sustainability 2020, 12, 6940. https://doi.org/10.3390/su12176940
and/or Natalucci, V.; Marini, C.F.; Flori, M.; Pietropaolo, F.; Lucertini, F.; Annibalini, G.; Vallorani, L.; Sisti, D.; Saltarelli, R.; Villarini, A.; Monaldi, S.; Barocci, S.; Catalano, V.; Rocchi, M.B.L.; Benelli, P.; Stocchi, V.; Barbieri, E.; Emili, R. Effects of a Home-Based Lifestyle Intervention Program on Cardiometabolic Health in Breast Cancer Survivors during the COVID-19 Lockdown. J. Clin. Med. 2021, 10, 2678. https://doi.org/10.3390/jcm10122678 etc.)
Please change and move the objectives explanation (Lines 80-91) at the beginning of the experimental design section, here can be misleading.
Experimental design:
Will the author include in the study patients with all types of surgery? Please include it in table 1.
Procedures:
3.5 3.6 and 3.6 are too fragmented, please rearrange the sections, most of the sections are assessments, so they can be categorized in one section called “assessments”.
Please do not repeat the objectives in this section.
Discussion:
I suggest to write a discussion, where a comparison with other online training is performed in order to understand why the protocol proposed by the authors should have a higher impact on compliance, functional, and clinical parameters.
Author Response
Reviewer 1
The title is misleading, as well as the entire article. The authors do not produce results about the feasibility of the study protocol proposed, but they explained in detail the procedure to perform a streaming web-based exercise performed at home, adapted to cancer patients. This information, especially after Covid-19 pandemic are very interesting and it is important to share this type of knowledge. I suggest to rewrite the introduction and adjust the objectives in order to this comment. It is clear that this is an ongoing study, but the aim should be the innovative protocol not the effect on these patients or the compliance, because there are no data yet and, in my opinion, it is not in line with the topic of the journal.
- Thank you for taking your time to review our protocol. We appreciate your feedback back and have considered your comments. The Methods and Protocol journal accepts innovative papers, as well as protocols without any data results. As a protocol paper, the reasoning behind the layout and descriptions provided follow the layout and standard of the journal requirements for protocol submissions.
Introduction:
Please include some background about the effect of virtual exercise on cancer patients ( i.e. Grazioli, E.; Cerulli, C.; Dimauro, I.; Moretti, E.; Murri, A.; Parisi, A. New Strategy of Home-Based Exercise during Pandemic COVID-19 in Breast Cancer Patients: A Case Study. Sustainability 2020, 12, 6940. https://doi.org/10.3390/su12176940
and/or Natalucci, V.; Marini, C.F.; Flori, M.; Pietropaolo, F.; Lucertini, F.; Annibalini, G.; Vallorani, L.; Sisti, D.; Saltarelli, R.; Villarini, A.; Monaldi, S.; Barocci, S.; Catalano, V.; Rocchi, M.B.L.; Benelli, P.; Stocchi, V.; Barbieri, E.; Emili, R. Effects of a Home-Based Lifestyle Intervention Program on Cardiometabolic Health in Breast Cancer Survivors during the COVID-19 Lockdown. J. Clin. Med. 2021, 10, 2678. https://doi.org/10.3390/jcm10122678 etc.)
- Thank you for the suggestions. These references have been added to the second paragraph of the introduction which are now lines 54-73.
Please change and move the objectives explanation (Lines 80-91) at the beginning of the experimental design section, here can be misleading.
- Thank you for the comments. Since we are submitting a protocol paper, we wanted to highlight endpoints as part of the study purpose.
Experimental design:
Will the author include in the study patients with all types of surgery? Please include it in table 1.
- Thank you for your feedback. We have updated Table 1 to include a statement about including patients with any type of surgery.
Procedures:
3.5 3.6 and 3.6 are too fragmented, please rearrange the sections, most of the sections are assessments, so they can be categorized in one section called “assessments”.
- Thank you for your suggestion. We have reframed the assessments to avoid so many subsections.
Please do not repeat the objectives in this section.
- Thank you for your comments. The endpoints are further explained in the this section to fully describe what we are measuring, thank you.
Discussion:
I suggest to write a discussion, where a comparison with other online training is performed in order to understand why the protocol proposed by the authors should have a higher impact on compliance, functional, and clinical parameters.
- Thank you for your suggestion. Based on manuscript requirements in the journal of Methods and Protocols, a discussion is not applicable when submitting a protocol to this journal. We will gladly include a discussion if the Editor of the journal feels we should provide one despite it not being part of the manuscript requirements.
Reviewer 2 Report
This methodological paper proposed a protocol to assess the effect of PA on cancer survivors. The proposed methodology is sound. Statistics is fine and overall the paper is acceptable for the publication.
The paper aim at suggesting a method to assess different exercise protocols in cancer survivors.
The topic is relevant for the field of exercise and cancer, exercise and stress management, quality of life.
The proposed methodology can be helpful for the clinic.
The proposed methodology is appropriate.
The conclusions consistent with the evidence and arguments presented they address the main question posed.
References are appropriate, albeit the reference list cite a few paper who evaluated different exercises protcols in cancer survivor. More references can be added concerning this point.
Table and figures are fine....
Author Response
Reviewer 2
This methodological paper proposed a protocol to assess the effect of PA on cancer survivors. The proposed methodology is sound. Statistics is fine and overall the paper is acceptable for the publication.
- Thank you for taking time to review our protocol. We appreciate the positive feedback and comments and have taken them into consideration.
The paper aim at suggesting a method to assess different exercise protocols in cancer survivors.
The topic is relevant for the field of exercise and cancer, exercise and stress management, quality of life.
The proposed methodology can be helpful for the clinic.
The proposed methodology is appropriate.
The conclusions consistent with the evidence and arguments presented they address the main question posed.
References are appropriate, albeit the reference list cite a few paper who evaluated different exercises protcols in cancer survivor. More references can be added concerning this point.
- Thank you for your feedback. More references have been added to the second paragraph of the introduction which are now lines 54-73.
Table and figures are fine....
Reviewer 3 Report
MPS-234205 presents a protocol for a physical activity intervention for cancer survivors. Given that this is a protocol, my comments are meant to improve the presentation instead of vet the procedures. I hope the authors consider my feedback.
· Lines 46-47: And because sedentary time may have been high prior to cancer diagnosis.
· Lines 74-75: Statement needs citation.
· Line 91: Delete last sentence.
· Line 111: Use plus-minus symbol instead. The same comment applies elsewhere in the paper.
· Table 1: Make sure that all abbreviations are defined in a table note so the table (and any other data element) can stand-alone. All data elements (tables and figures) need to stand-alone.
· Section 3.5.2: Consider the PAR-Q+ instead.
· Make any changes to the abstract that align with those from the text.
Author Response
Reviewer 3
MPS-234205 presents a protocol for a physical activity intervention for cancer survivors. Given that this is a protocol, my comments are meant to improve the presentation instead of vet the procedures. I hope the authors consider my feedback.
We appreciate your time and effort in reviewing our protocol. We have updated our protocol to reflect your suggestions, thank you.
- Lines 46-47: And because sedentary time may have been high prior to cancer diagnosis.
Thank you for your feedback. Line 46-47 (now lines 47-48) have been adjusted to reflect your suggestion.
- Lines 74-75: Statement needs citation.
A citation has been added for lines 74-75 (now lines 76-77), thank you.
- Line 91: Delete last sentence.
Thank you for your feedback. Line 91 (now line 93) has been deleted based off your suggestion.
- Line 111: Use plus-minus symbol instead. The same comment applies elsewhere in the paper.
Thank you. The figure description has been adjusted accordingly, along with the rest of the document.
- Table 1: Make sure that all abbreviations are defined in a table note so the table (and any other data element) can stand-alone. All data elements (tables and figures) need to stand-alone.
Thank you for the suggestions. Table 1, as well as the rest of the document, have been adjusted to define any abbreviations.
- Section 3.5.2: Consider the PAR-Q+ instead.
Thank you for the suggestions. We have considered the PARQ+ and will utilize the questionnaire for our study.
- Make any changes to the abstract that align with those from the text.
The abstract aligns with the current text, thank you.
Round 2
Reviewer 1 Report
Dear Authors, the manuscript is suitable for publication according to the journal standards. Best regards.